# Evaluating the Anticarcinogenic Activity of Surface Modified/Functionalized Nanochitosan: The Emerging Trends and Endeavors

**DOI:** 10.3390/polym13183138

**Published:** 2021-09-17

**Authors:** Jae-Wook Oh, Juhyun Shin, Sechul Chun, Manikandan Muthu, Judy Gopal

**Affiliations:** 1Department of Stem Cell and Regenerative Biotechnology, Konkuk University, Seoul 143-701, Korea; ohjw@konkuk.ac.kr (J.-W.O.); junejhs@konkuk.ac.kr (J.S.); 2Department of Environmental Health Science, Konkuk University, Seoul 143-701, Korea; scchun@konkuk.ac.kr (S.C.); bhagatmani@gmail.com (M.M.)

**Keywords:** chitosan, nanochitosan, sea food waste, biomedical applications, functionalization, surface modifications

## Abstract

Chitosan begins its humble journey from marine food shell wastes and ends up as a versatile nutraceutical. This review focuses on briefly discussing the antioxidant activity of chitosan and retrospecting the accomplishments of chitosan nanoparticles as an anticarcinogen. The various modified/functionalized/encapsulated chitosan nanoparticles and nanoforms have been listed and their biomedical deliverables presented. The anticancer accomplishments of chitosan and its modified composites have been reviewed and presented. The future of surface modified chitosan and the lacunae in the current research focus have been discussed as future perspective. This review puts forth the urge to expand the scientific curiosity towards attempting a variety of functionalization and surface modifications to chitosan. There are few well known modifications and functionalization that benefit biomedical applications that have been proven for other systems. Being a biodegradable, biocompatible polymer, chitosan-based nanomaterials are an attractive option for medical applications. Therefore, maximizing expansion of its bioactive properties are explored. The need for applying the ideal functionalization that will significantly promote the anticancer contributions of chitosan nanomaterials has also been stressed.

## 1. Introduction

The biopolysaccharide chitosan is recovered from chitin through the deacetylation process. Chitin, the primary structural polymer occurring in the exoskeletons of crustaceans, mollusks, and insects [1,2,3,4], is second to cellulose, which is the first naturally abundant polysaccharide [1]. Chitosan is derived from chitin following alkaline deacetylation, leading to a structure having 2-amino-2-dedoxy-d-glucose and 2-acetamino-dedoxy-d-glucose units linked by β-(1→4) bonds [1,2]. Amino groups are the principal functional groups of chitosan. The degree of deacetylation (DDA) is opposed to the degree of acetylation (DA), DA = 100-DDA or DDA = 100-DA. DA is also very important for solubility and in separating the terms chitin and chitosan: if DA > 50%, the biopolymer is chitin-like, if DA < 50%, the biopolymer is chitosan-like

Generally, chitosan has three types of reactive functional groups [5]. Chitosan has an amino group at the C2 position of each deacetylated unit and hydroxyl groups at the C6 and C3 positions. These play a significant role while modifying chitosan for enhancing its biological activity. The amine at the C-2″ position is the one responsible for the biological activity of chitosan [6]. Chitosan has received significant attention for accomplishing several biological activities. The degree of deacetylation (DDA) has an important say on the bioactivity of chitosan [3,4,7,8,9,10].

Chitosan and its derivatives have been extensively applied into medical and pharmaceutical applications (Figure 1). This is because of their highly competitive biological properties that include: biocompatibility, biodegradability, hypocholesterolemic, antimicrobial, nontoxic and antitumor, analgesic, hemostatic, and antioxidant properties [11,12]. These properties promote chitosan as an ideal candidate for biomedical applications, wound healing, tissue engineering, and for drug and gene delivery [13,14,15,16,17]. Chitosan morphologies include: films, gels, membranes, nanoparticles, coatings, suspensions, and hydrogels. Each of these unique morphologies are capable of influencing their biomedical activities and properties [18]. 

Because of its status ‘generally recognition as safe’ (GRAS) and its excellent biodegradability, chitosan has been extensively employed for the encapsulation of bioactive compounds [19]. The ability of chitosan towards the loading and delivery of sensitive bioactive compounds [20,21], polyphenolic compounds [19,22], and vitamins [23] has been well documented. The functional amino groups and their flexibility to modifications enhance their mechanical and physical properties. Because of chitosan’s high molecular weight, it shows limited bioavailability. Depolymerization by hydrolysis of polymer chains helps acquire low molecular oligomers of chitosan [24]. The enzymatic (using lysosome, chitinase, pectinase, cellulose) degradation of chitosan is also gaining attention [23,25]. Pepsin, papain, pronase [26,27], hepatopancreas [28], and chitosanase [25] can yield chitosan of low molecular weight [3]. 

Aranaz et al. reported that when the DDA increases, chitosan’s solubility also increases and biological interactions also increase [3]. The cationic characteristic of chitosan promotes its interaction with proteins, phospholipids, therapeutic DNA or RNA, bile, fatty acids, and anionic polyelectrolytes [11,13,14,29]. The high viscosity and low solubility of chitosan limits its biological applications [18].

The following review summarizes the antioxidant milestones reached through engaging chitosan for biomedical applications. The importance of nanochitosan and its allies, ideally in areas of biomedical applications, has been extensively reported. The reports on the antioxidant activity of chitosan have been briefly discussed and duly acknowledged. Surface modification of nanochitosan and specific functionalization that has aided in attaining significant medical milestones have been discussed. The need for furtherance of more such nanochitosan modifications and their prospective candidature in expanding its bio-applicability has been speculated and discussed as a future perspective. 

## 2. Antioxidant Activity of Chitosan

Much research work has been directed in the area of free radical reactions, since they have been held responsible for several specific human diseases. Reactive oxygen species (ROS) are formed within the human body, in the process of normal metabolism. ROS are highly interactive with biomolecules and they oxidize lipids, proteins, carbohydrates, and DNA, ultimately leading to oxidative stress [2]. Enzymes such as catalase, superoxide dismutase and glutathione peroxidase are the innate cellular defense systems available against ROS-mediated cellular injury [30]. On excessive generation of ROS, the defense mechanism becomes inadequate and leads to oxidative stress. Oxidative stress is now associated with hypertension, dyslipidemia aging, rheumatoid arthritis, cancer, myocardial infraction, atherosclerosis, heart failure, angina pectoris, wrinkle formation, inflammation, and neurodegenerative diseases such as Alzheimer, Parkinson, and amyotrophic lateral sclerosis [31,32,33,34]. 

As a remedy to the ROS issue, the search for antioxidants intensified. It is in this context, that the antioxidant activity of chitosan has become rather attractive. Chitosan is an attractive option since it is a biopolymer and it is economical, biodegradable, and versatile. Chitosan has demonstrated notable scavenging activity against different radical species. Xie et al. proposed several theories on the scavenging activity of chitosan and its derivatives against free radicals [35]: (i) The hydroxyl groups in chitosan react with hydroxyl radicals via H-abstraction reaction. (ii) OH reacts with residual-free amino NH_2_ to form stable macromolecules radicals. (iii) The NH_2_ groups absorb H^+^ from the solution to form ammonium groups and react with OH through other addition reactions. These are the plausible mechanisms deciphered for the antioxidant strategies of chitosan [35].

The ROS scavenging capacity of chitosan largely rests on the DDA and MW of chitosan [3]. Chitin being insoluble in water is limited from being used as an antioxidant agent. The NH_2_ groups of chitosan deal with the scavenging activity and they can be protonated in an acidic solution. Samar et al. assessed the antioxidant activity of different DDA and MW chitosan samples. They concluded that higher DDA and lower MW are characteristic patterns for higher antioxidant activity [9]. Hajji et al. studied antioxidant activity of chitosan from crab (*Carcinus mediterraneus*) shells (DDA: 83%), Tunisian marine sources shrimp (*Penaeus kerathurus*) waste (DDA: 88%), and cuttlefish (*Sepia officinalis*) bones (DDA: 95%) [36]. Chitosan from cuttlefish showed highest antioxidant potential. Kim and Thomas proved that higher antioxidant activity was obtained with low MW chitosan (30 kDa) [37]. 

Sun et al. evaluated the scavenging capacity of chitosan oligomers with different MW, against hydroxyl radicals and superoxide anion [38]. In both superoxide anion and hydroxyl radical, lower MW worked well. Chang et al. reported antioxidant activity of chitosan against hydrogen peroxide, 2,2-diphenyl-1-picrylhydrazyl (DPPH) radical, and chelating ferrous ion [39]. The results confirmed that low MW chitosan (~2.2 kDa) has the highest impact. Li et al. confirmed that the MW of chitosan and its concentration are major attributes [40]. 

Although the antioxidant activity of chitosan is strongly proven, the lack of an H-atom donor to serve as a good chain-breaking antioxidant is a key limitation [41]. The scavenging capacity of free radicals is correlated with the O–H or N–H bond dissociation energy and the stability of the radicals. Because the intramolecular and intermolecular hydrogen bonds in chitosan molecules are strong, the OH and NH_2_ groups are hard to dissociate [35]. This is the reason why various surface modifications and functionalization of chitosan were sought after. Chitosan molecules were modified by grafting functional groups into their molecular structure. We will briefly discuss the existing modifications of chitosan, for its antioxidant activity as well as for serving enhanced bioactivities.

## 3. Surface Modifications of Chitosan

Surface modifications are ways by which superior properties replace weaknesses of the core material through modifying the surfaces physically/chemically. In case of chitosan, its backbone is modified to alter properties such as solubility, mucoadhesion, and stability. Ideally, the -NH_2_ and -OH groups are the active modification sites. Chitosan polymers are modified through: blending, graft co-polymerization, and curing [42]. The mixing of two or more polymers involves blending, which is a form of physical modification. This is the simplest form of surface modification. Few examples of surface modification of chitosan by blending are poly (vinyl alcohol) (PVA), poly (vinyl pyrrolidone) (PVP), and poly (ethyl oxide) (PEO). PVA modification of chitosan improves the tensile strength and water vapor permeability of chitosan films [43]. PVA-chitosan also show improved mechanical properties for controlled drug delivery [44]. Amoxicillin formulated with a crosslinked chitosan/PVP blend with glutaraldehyde to form a semi-interpenetrating polymer network (semi-IPN) is an example of physical modification [45]. 

Graft co-polymerization is the covalent bonding of polymers. Curing converts, the polymers into a solidified mass by means of thermal, electrochemical, or ultraviolet radiation [46]. Chemical modification is achieved by altering the functional groups in a compound by chemical, radiation, photochemical, plasma-induced, and enzymatic grafting methods [42]. Chemical modification of chitosan results in quaternized chitosan, thiolated chitosan, carboxylated chitosan, amphiphilic chitosan, chitosan with chelating agents, PEGylated chitosan, and lactose-modified chitosan. The primary amine (-NH_2_) group is the chemical modification site, key to pharmaceutical applications [42], reacting with sulphates, citrates, and phosphates [47]. This enhances the stability and drug encapsulation efficiency of the modified chitosan [48]. Improved solubility of chitosan in intestinal media is achieved through N-trimethyl chitosan chloride (TMC), a quaternized chitosan [49]. The mucoadhesiveness of chitosan is enhanced incorporating thiolated chitosan [50]. Quaternization of chitosan forms several derivatives such as dimethylethyl (DMEC), diethylmethyl (DEMC), trimethyl (TMC), and triethyl chitosan (TEC). Quaternization of chitosan helps in increasing the permeability of insulin across Caco-2 cells [51]. Chitosan-thioglycolic acid, chitosan-cysteine, chitosan-glutathione, and chitosan-thioethylamidine are some of the thiolated chitosan derivatives presently in use. Grafting of polyphenols on chitosan has been studied [52,53,54,55]. Phenolic compounds are oxidized to o-quinones and further covalently grafted to nucleophilic amine groups (through Schiff-base and/or Michael-type addition reaction) [52,56]. Chitosan modified by grafting polyphenols, showed significantly enhanced antioxidant activity, a classic review of the above-mentioned modifications has been published [57]. 

## 4. Nanochitosan and Its Modifications

### 4.1. Methods Used to Prepare Nanochitosan

Nanotechnology has impacted almost every area of science and technology. Nanomaterials are now accepted to be highly versatile, and superior compared to their bulk counterparts. Hence, the nanoforms of almost all successful materials are readily attempted, since they are destined and proven to have a higher success rate. In case of chitosan, formulation of chitosan nanoparticles and nanomaterials has been attempted. Chitosan nanomaterials are synthesized using emulsification, emulsion based solvent evaporation, ionotropic gelation, microemulsion, and solvent diffusion. Drugs are loaded within the chitosan nanomaterials via electrostatic interaction, hydrogen bonding, and hydrophobic interactions. Coacervation is done by separating spherical particles by mixing electrostatically driven liquids [58,59]. In the polyelectrolyte complex (PEC) method, an anionic solution is added to the cationic polymer under mechanical stirring, to obtain nanoparticles [60,61]. In the coprecipitation method, chitosan solution is added, leading to coprecipitation of highly monodisperses chitosan nanoparticles [62]. In the microemulsion method, chitosan in acetic acid solution and glutaraldehyde are added to a surfactant in an organic solvent such as hexane for cross-linking [63]. Emulsification Solvent Diffusion Method is where an o/w emulsion is prepared under mechanical stirring followed by high pressure homogenization [64,65]. Emulsion Based Solvent Evaporation Method is a slight modification of the above method that uses no high shear forces. Reverse Micellar Method is where a surfactant is added to an organic solvent followed by the addition of chitosan, drug, and crosslinking agent [66]. 

### 4.2. Chitosan/Functionalized Chitosan Nanocarriers—A Snap Shot of Biomedical Achievements

Chitosan has been widely used to produce nanoparticles as standalone materials or in combination with others. In addition to this, chitosan nanomaterial has been functionalized with various bioactive moieties in order to enhance its biological properties and overcome the limitations of the source material. Surface modification is the act of modifying the surface of a material by altering the surface of the material to impact their original physical, chemical, or biological characteristics. Surface functionalization introduces chemical functional groups to a surface. The onset of nanotechnology has brought about key advances in nanomaterial-based applications. Functionalization of nanoparticles has overcome default properties of the nanomaterial and brought about improvisation. This is believed to hold high promises towards pharmaceutical and biomedical sciences. Functionalization of nanoparticles has emerged as a highly promising routine to yield multifunctional nanoparticles that have overcome inherent weakness. 

Functionalization of chitosan nanoparticles and their surface modification especially to suit their therapeutic applications has been successfully demonstrated. We present a brief overview of the reports on the diverse modified/functionalized chitosan nanoparticles and their applications. Chitosan has been formulated as polymeric nanoparticles for oral drug delivery by conjugating antioxidants, catechin and epigallocatechin (flavonoids from green tea) with chitosan nanoparticles. Usually, these catechins are poorly absorbed across intestinal membranes and undergo degradation in intestinal fluid. This is overcome by encapsulating these catechins inside chitosan nanoparticles [67]. Tamoxifen, a water soluble anti-cancer drug, is useful for oral cancer. Tamoxifen encapsulated in lecithin-chitosan nanoparticles led to the successful movement of tamoxifen across the intestinal epithelium [68]. Nanoparticles of doxorubicin hydrochloride (DOX)/chitosan/carboxymethyl chitosan, increased the intestinal absorption of DOX [69]. Alendronate sodium (drug for osteoporosis) suffers from gastrointestinal side effects and low oral bioavailability. Encapsulation of alendronate sodium in chitosan nanoparticles overcame the above mentioned limitations. Sustained drug release of sunitinib (a tyrosine kinase inhibitor) was achieved by encapsulating the drug in chitosan nanoparticles [70]. Insulin-loaded chitosan nanoparticles were functionalized with tripolyphosphate (TPP), which enhanced the NP uptake by the stomach epithelium [71]. Bay 41-4109, an active hepatitis B virus inhibitor, was formulated into chitosan nanoparticles improving drug solubility and oral bioavailability [72]. 

One of the advantages of using chitosan is its mucoadhesive property, this makes chitosan a successful candidate for nasal and intestinal drug delivery [73]. Carboxymethyl chitosan nanoparticles of carbamazepine, enhanced their bioavailability and brain targeting through the nasal passage [74]. Thiolated-chitosan nanoparticles of the drug leuprolide showed significant enhancement in the transportation of the leuprolide drug across porcine nasal mucosa. A nanoparticle based dry powder inhalation (DPI) of rifampicin (antitubercular drug) was formulated with chitosan as the polymer, achieving sustained drug release upto 24 h and zero toxicity [75]. 

Itraconazole, an anti-fungal drug suffers low solubility during oral adminstration. The drug in spray-dried chitosan nanoparticles alongwith lactose, mannitol, and leucine, increased deposition of itraconazole in the lungs [76]. Giovino et al. developed chitosan buccal films of insulin loaded poly (ethylene glycol) methyl ether-block-polylactide (PEG-b-PLA) nanoparticles [77], that exhibited excellent mucoadhesive properties and sustained insulin release. Curcumin prepared as polycaprolactone nanoparticles coated with chitosan have been used for buccal delivery. Nanochitosan encapsulated enriched flavonoid fraction (EFF-Cg) loaded PLGA nanoparticles were used for buccal delivery as chitosan films. The bioavailability of EFF-Cg was improved with no cytotoxicity [78]. 

Nayak et al. prepared chitosan nanoformulations using ascorbic acid (Vitamin C), α-tochopherol (Vitamin E), and catechol, along with green synthesized AgNPs for targeted drug delivery to breast cancer cells. These nanoformulations possessed higher antioxidant activity and also were hemocompatible [79]. Curcumin encapsulated by chitosan-tripolyphosphate (CS-TPP) nanoparticles were demonstrated to show high radical scavenging activity [80]. In another study, Kaur et al. [81] formulated catechin hydrate (CH)-loaded nanoparticles functionalized with TPP. These catechin loaded nanoparticles showed higher antioxidant activity. Chlorogenic acid (CGA), a polyphenolic antioxidant, was encapsulated into chitosan nanoparticles and demonstrated for their improved antioxidant activity [82]. Chitosan/DNA co-assemblies were used for the encapsulation of astaxanthin, a carotenoid that possesses strong antioxidant properties [77,83].

Chitosan-vancomycin nanoparticles for colon delivery were prepared for better drug release [84]. Coco et al. have studied the comparative ability of chitosan nanoparticles against other polymers, for inflamed colon drug delivery [85]. Trimethyl nanochitosan entrapping ovalbumin (OVA) showed high permeability of OVA. Chitosan-carboxymethyl starch nanoparticles of 5-aminosalicylic acid were demonstrated for inflammatory bowel disease, showing controlled drug release [85]. Rosmarinic acid loaded chitosan nanoparticles have been used for non-cytotoxic ocular delivery. Imiquimod was prepared as chitosan coated PCL nanocapsules embedded in hydroethylcellulose gel and also as PCL nanocapsules embedded in chitosan hydrogel for vaginal delivery for treating human papillomavirus infection [86]. The former showed higher mucoadhesion, and the latter higher drug permeation. 

Chitosan-based material have been used as bone substitutes, due to their excellent biocompatibility and biodegradability. However, the hydrophobic surface of chitosan films inhibits osteogenesis mineralization process during bone regeneration. To resolve this issue, a novel polydopamine-modified chitosan film with good hydrophilicity has been developed for bone tissue engineering applications. These films showed enhanced growth rate of apatite on the modified chitosan film. Due to the method being capable of generating large quantities in bulk, this sure has a huge potential in the area of bone tissue engineering. Table 1 lists the various modified chitosan nanomaterials with their biomedical applications. 

## 5. Anticarcinogenic/Antitumour Activity of Chitosan Nanomaterials

### 5.1. Chitosan Nanocarriers—Anticancer Impacts

Chitosan nanocarriers have also been reported for their accomplishments in cancer research. Nanochitosan have raised the impact of the chitosan polymer further through nanostructurization. We summarize the available reports confirming the use of nanochitosan as nanocarriers for cancer therapy. Mifepristone (MIF) is an anticancer drug used against various cancers, Zhang et al. [142] developed MIF-loaded chitosan nanoparticles (MCNs) that exhibited increased anticancer activity in several cancer cell lines. Pharmacokinetic studies in male rats orally administered with MCNs showed a 3.2-fold increase compared to free MIF. Wang et al. [143] designed chitosan nanoparticles for co-delivery of 5-fluororacil and aspirin and induced synergistic antitumor activity through playing around with nuclear factor kappa B (NF-κB)/cyclooxygenase-2 (COX-2) signaling pathways. The designed chitosan nanocarrier operated via aspirin-induced suppression of NF-κB and inhibition of COX-2.

The anti-metabolic compounds pyrazolopyrimidine and pyrazolopyridine thioglycosides were synthesized and encapsulated by chitosan nanoparticles. Their cytotoxicity against Huh-7 and Mcf-7 cells, related to liver and breast cancer cells, was successfully demonstrated [144]. Deepa et al. demonstrated the successful release of cytarabine against solid tumors, using nanochitosan formulations [145]. Cavalli et al. prepared chitosan nanospheres with 5-FU, which were effective in reducing tumor cell proliferation and were able to inhibit both HT29 and PC-3 adhesion to HUVEC [146]. Sahu et al. prepared 5-FU loaded biocompatible chitosan nanogels (FCNGL) that released 5-FU in an acidic environment, resulting in selective drug delivery, leading to sustained delivery of 5-FU for chemotherapy. This enabled high efficacy, patient compliance, and safety [147]. Keerthikumarc et al. synthesized chitosan encapsulated curcumin nanoparticles showing sustained release of the drug and high anticancer efficacy in human oral cancer cell lines [148]. Shahiwala et al. synthesized chitosan nanoparticles in an alcoholic extract of *Indigofera intricate* and demonstrated 500-fold reduction in the extract concentration, when the chitosan nanocarriers laden with plant extracts were used [149]. Alipour et al. demonstrated the sustained release of silibinin-loaded chitosan nanoparticles (SCNP) against C6 glioma cells [150]. In another study [151], the effects of nanochitosan on tumor growth were investigated using nude mice xenografted with human hepatocellular carcinoma (HCC) (BEL-7402) cells. The results demonstrated that the treatment of these nude mice with nanochitosan significantly inhibited tumor growth and induced tumor necrosis. 

### 5.2. Surface Modified/Functionalized Chitosan Nanocarriers—Anticarcinogenic Impacts

Chitosan modifications and functionalization have also been demonstrated for their anticancer activity. Chitosan based nanoparticles of Bay 41-4109 showed prolonged circulation in the blood and enhanced intestinal absorption [72]. Enoxaparin has less oral bioavailability; this was overcome by enoxaparin-loaded alginate-coated chitosan NPs (Enx-Alg-CS-NPs) and demonstrated using rat models. Novel hepatocyte-targeted delivery system with glycyrrhizin (GL) modified N-caproyl chitosan (CCS) was demonstrated in rats. These CCS-NPs-GL were demonstrated in these rate models to be able to bring about effective drug delivery for hepatocyte targeting. Sharifi-Rad et al. have elaborately published a review on chitosan nanoparticles as a promising tool in nanomedicine with particular emphasis on oncological treatment [152].

Chitosan modified mesoporous silica nanoparticles (MSN) offer high surface area and pore volume, including stability of chitosan at different pH values. Controlled release profile of the curcumin drug molecule has been demonstrated [153]. Amphiphilic chitosan derivatives (N-octyl-N-mPEG-chitosan, mPEG = poly(ethyl-ene glycol) monomethyl ether; OPEGC) showing good water solubility and low cytotoxicity were successfully synthesized via the Schiff base reduction reaction of chitosan [154]. Copper-loaded nanochitosan were prepared for the effective treatment of osteosarcoma [155]. The copper-loaded chitosan nanoparticles (CuCNPs) exhibited remarkable anticancer activity. The superior anticancer effect of CuCNPs is attributed to the generation of a higher mitochondrial ROS level compared to that of the control. Overall, the anticancer effect of copper has been enhanced by delivering it within biocompatible nanochitosan.

Methotrexate (MTX) has poor water solubility, low bioavailability, and leading to resistance in cancer cells. Novel folate redox-responsive chitosan (FTC) nanoparticles for intracellular MTX delivery helped confer redox responsiveness and active targeting of folate receptors (FRs) [156]. These possess tumor specificity and controlled drug release due to the overexpression of FRs and high concentration of reductive agents in the cancer microenvironment. FTC-NPs showed better inhibitory effects on HeLa cancer cells compared to non-target chitosan-based NPs. Methotrexate (MTX) and mitomycin C (MMC) loaded PEGylated chitosan nanoparticles (CS-NPs) were developed as drug delivery systems [157]. MTX, as a folic acid analogue, was employed as a tumor-targeting ligand. Effective uptake via FA receptor-mediated endocytosis and codelivery of MTX and MMC at the tumor site have been reported. (MTX + MMC)-PEG-CS-NPs as targeted drug codelivery systems can have clinical implications for combinational cancer chemotherapy. Irinotecan nanoparticles (NPs) using folate–chitosan conjugate (FCC) for more effective delivery of Irinotecan for killing breast cancer cells was developed [158]. Since breast cancer cells express folate receptors on their surface, these irinotecan-loaded folic acid–chitosan conjugated nanocarriers could be used for targeted delivery against metastatic breast cancer with some modifications.

Liu et al. prepared chitosan grafted halloysite nanotubes (HNTs-g-CS) as potential nanocarriers for drug delivery in cancer therapy, as curcumin loaded HNTs-g-CS increased apoptosis on EJ cells [159]. Abbas et al. introduced a chitosan (CS) and CS magnetic nanoparticles (MNPs) encapsulating polyvinylpyrrolidone (PVP)/maltodextrin (MD)-based microparticles (MPs) system that was inhalable delivering the drug to deep lung tissues [160]. Almutairi et al. prepared raloxifene-encapsulated hyaluronic acid-decorated chitosan nanoparticles that showed cytotoxicity against human lung A549 cancer cell lines [161]. Bae et al. prepared self-aggregates from deoxycholic acid-modified chitosan for use as delivery vehicles of anticancer drugs [162]. Wu et al. synthesized 10-hydroxycamptothecine nanoneedles integrated with an exterior thin layer of the methotrexate-chitosan conjugates, for enhanced therapeutic performances in cancer treatment [163]. Li et al. synthesized the drug carrier (Fe_3_O_4_/carboxymethyl-chitosan nanoparticles) and demonstrated with the antitumour drug rapamycin (Fe_3_O_4_/CMCS-Rapa NPs) [164]. Roy et al. encapsulated Fe_3_O_4_-bLf (Fe_3_O_4_-saturated lactoferrin) in alginate enclosed chitosan-coated calcium phosphate (AEC-CP) nanocarriers (NCs) to be usesd against tumor in mice [165]. Arunkumar et al. synthesized composite injectable chitosan gel (DZ-CGs) comprising of doxorubicin-loaded zein nanoparticles (DOX-SC ZNPs), which could bring about successful in vitro drug release in a controlled manner. The composite DZ-CGs were more effective in killing cancer cells [166]. Hwang et al. synthesized the hydrophobically modified glycol chitosan (HGC) nanoparticles loaded with the anticancer drug docetaxel (DTX), leading to reduced tumor volume of A549 lung cancer cells [167].

Gomathi et al. prepared letrozole with chitosan nanoparticles using sodium tripolyphosphate as the crosslinking agent for anticancer treatment. This was biocompatible and possessed hemocompatible properties, which makes it an efficient nanocarrier for the anticancer drug letrozole [168]. Wang and Zhao optimized the preparation of anticancer drug—gefitinib with chitosan protamine nanoparticles [169]. Koo et al. reported the preparation of water-insoluble paclitaxel encapsulated into glycol chitosan nanoparticles with hydrotropic oligomers (HO-CNPs), these paclitaxel-HO-CNPs showed higher therapeutic efficacy than the commercial Abraxane^®^ formulation [170]. Maya et al. prepared O-carboxymethyl chitosan (O-CMC) nanoparticles, surface-conjugated with cetuximab (Cet) for targeted delivery of paclitaxel. These can be used for targeted therapy of epidermal growth factor receptor (EGFR) in overexpressing cancers [171]. Al-Musawi et al. synthesized chitosan-covered superparamagnetic iron oxide nanoparticles (CS-SPION) and applied them as a nano-carrier for loading of (5-FU) (CS-5-FU-SPION) [172]. Anitha et al. prepared a nanoformulation of curcumin using dextran sulphate and chitosan, leading to preferential killing of cancer cells compared to normal cells by the curcumin-loaded drug [173]. Baghbani et al. prepared curcumin-loaded chitosan/perfiuorohexane nanodroplets using a nanoemulsion process [174]. Rajan et al. synthesized curcumin nanoparticles loaded in chitosan biopolymer and bovine serum albumin, which resulted in selective drug targeting of colorectal carcinoma cells [175]. George et al. reported the preparation of functionalized nanohybrid hydrogel using l-histidine (HIS) conjugated chitosan, phyto-synthesised zinc oxide nanoparticles (ZNPs) and dialdehyde cellulose (DAC) for sustained drug delivery of naringenin, quercetin, and curcumin. Anticancer studies towards A431 cells (epidermoid carcinoma) exhibited excellent cytotoxicity, compared to the free polyphenol drugs [176]. Chaichanasak et al. prepared chitosan-based nanoparticles with damnacanthal (DAM), leading to improved anticancer effects [177]. Oh et al. have elaborated on the various medical and drug delivery applications of chitosan and green synthesized chitosan nanomaterials in previously published reviews [178,179].

Although the exact mechanism behind the anticancer activity of chitosan remains elusive, Adhikari and Yadav (2018) have elucidated few plausible mechanisms that may explain the mechanism involved in the anticancer activity of chitosan. Table 2 presents a consolidated list of the mechanisms suggested for chitosan and nanochitosan mode of anticarcinogenic activity. Those included are: (i) permeation enhancing mechanism, (ii) antiangiogenic mechanism, and (iii) sustained release mechanism [180] (Figure 2). Chitosan has been demonstrated for its anticarcinogenic effect on MDA-MB-231 [181,182] and antiproliferative effect on T24 urinary bladder cell lines [183,184], while chitosan nanoparticles have been proven for their antiangiogenic effect on human hepatocarcinoma [151,185] and cell proliferation inhibition on BEL7402, HT-29 cell lines [186,187,188,189,190,191,192]. Mifepristone (MIF) loaded chitosan nanoparticles have demonstrated an enhanced anticarcinogenic effect through sustained drug release as well as enhanced bioavailability of MIF [142].

## 6. Limitations and Future Endeavors

### 6.1. Toxicity Aspects of Chitosan

Chitin has its own list of biomedical applications; chitosan has its own credentials far surpassing those of chitin. Nanochitosan has also made significant progress. Chitosan nanoparticles effectively deliver drugs to the specific sites by retaining the drug longer, allowing extended time for drug absorption. As in the case of any material used for biomedical applications, the toxicity aspects are quiet a concern. Chitosan is biodegradable and its degradation is dependent on the degree of deacetylation and the availability of amino groups. The toxicity of chitosan increases as charge density increases and degree of deacetylation increases too [191]. As of now, no human toxicity reports on chitosan-based formulations exist, but there are several animal toxicity-reports on its safety in vivo and in vitro. Aluani et al. have studied and confirmed the in vivo toxicity of two types of quercetin-loaded chitosan NPs (QR-NP1, QR-NP2) on male Wistar albino rats [192]. Their data concluded that chitosan nanoparticle are safe carriers for quercetin in oxidative stress associated injuries. Death and malformation of zebrafish embryos occurred with increasing chitosan nanoparticle concentrations. Almost 100% mortality was observed at a concentration of 40 mg/L for the 200 nm chitosan nanoparticles. Therefore, chitosan toxicity appears to be dose-dependent and needs to be considered at high concentrations [193]. Toxicity studies of chitosan and chitosan nanoparticles is lacking; it is necessary that chitosan should not be taken for granted in terms of its toxicity. When it specially comes to chitosan nanoforms, it is essential that well defined toxicity assessment is made for each and every study, with respect to specific cell lines and their relevant experimental set ups. Nanomaterial properties are very different from their bulk and so toxicity aspects need to be studied for each respective system. In fact, it is necessary that any application reporting the use of chitosan nanocarriers for medical purposes should culminate in toxicity assessment and toxicity validation of their nanocarrier system. This review points out to this gap and the need to fill in. 

Clinical use of oral or mucoadhesive drug formulations containing chitosan are yet to be activated, however, human vaccines have been formulated that use chitosan as an adjuvant [194,195]. Novel chitosan-modified polylactic-co-glycolicacid nanoparticles (CS@PLGA nanoparticles) were formulated for improving the bioavailability of tolbutamide (TOL) [92]. Using mice models, Bronchial Calu-3 and alveolar 549 cells were used to study the effect of chitosan-based drugs targeted for drug delivery to lungs [196]. Chitosan-coated PLGA nanoparticles showed better biodistribution and lower toxicity compared to those without the chitosan coating [197]. Grenha et al. 2007 reported absence of toxicity in vitro using Calu-3 cells and A549 epithelial cells at specific concentrations [198]. In vitro cytotoxicity of chitosan nanoparticles against buccal cells (TR146) was evaluated by Pistone et al. [110]. Chitosan nanoparticles were less cytotoxic than alginate and pectin nanoparticles. Moreover, bulk chitosan was more cytotoxic than nanochitosan, because of the linker attached to chitosan nanoparticles. It was also observed that the cytotoxicity of chitosan nanoparticles was shown to be further reduced by increasing the concentration of the linker (TPP) or using chitosan with lesser degree of deacetylation. To date, there are almost no toxicity reports in animal models and no reports of major adverse effects in healthy human volunteers and clinical data are lacking. Even though chitosan is approved in dietary use, wound dressing applications and cartilage formulations, yet a chitosan-based drug formulations have not been approved for mass marketing [199].

### 6.2. Inadequate Clinical Testings

In terms of clinical studies, except for scattered one or two reports, no progress has been made. Without putting the formulations to clinical trials, we will not get a clear picture of its performance. This is a huge lacuna that this review points out, and one that needs to be addressed specifically in order to see progress in this area of research. A chitosan-based nasal formulation of morphine (Rylomine^TM^) is currently in Phase 2 clinical trials (UK and EU) and Phase 3 clinical trials in the U.S. When it reaches the market, it will pave way for similar products in the near future, as well as assist in discerning any unanticipated effects in humans [200]. A need for promoting the use of chitosan nanoparticles, in targeted cancer theranostics, dermatologic applications, and targeted parenteral drug delivery systems is also stressed [201,202,203]; apart from a few reports no concrete progress has been made in this direction. We hope that future work on chitosan nanoparticles will also focus on toxicity studies in humans.

### 6.3. Unexplored Arenas

In addition to the above concerns, this review points out the fact that almost 99% of the nanochitosan work is surrounding chitosan nanoparticles alone. While chitosan nanomaterial includes quantum dots, micelles, nanogels, nanofibers, nanowhiskers, and nanospheres, none of these advanced nanoforms have been involved as much as chitosan nanoparticles have been involved. As is well known, morphological impacts of nanomaterial could be distinct and unique. Given this fact, it is rather intriguing why none of these have been studied stand-alone or conjugated with the diverse modifications and functionalizations that chitosan nanoparticles have been studied with. This is an area worth addressing and applying for achieving more prominent deliverables. This review prompts progress and inputs in this direction. Moreover, not much has been evaluated with respect to enhancement of antioxidant properties of chitosan nanoparticles as yet. Antioxidant activity of chitosan is one of the outstanding features of chitosan that validates its use in biomedical applications. With this being the case, nanoantioxidant chitosan should have hit the headlines by overcoming limitations. Compared to bulk materials, nanomaterials have always outshined and broken barriers. In this case too, such positive reports were expected and not much evidenced. We hope this review will trigger enthusiasm in this direction too. Extensive reviews pertaining to drug delivery applications of chitosan nanoparticles have been published by Li et al., 2018 [204] and Mohammed et al., 2017 [153], a lot less has been achieved in other areas of biomedical applications, such as tissue engineering, bone/skin grafts, antitumor applications, and the like. This review encourages expansion under these themes. 

As much as encapsulation, surface modifications of chitosan are reported, a lot less on actual functionalization with usual functional moieties that are prevalently used for biological functions, have been reported. PLA, PVA, and PEG are repetitively used, when there is a school of other groups that could prove worthy of assessment. Moreover, a lot fewer less reports on functionalization of well-known biocompatible inorganic nanomaterial such as Au, iron oxides, TiO_2_, Ag, C, Pt, Pd, etc., with chitosan exist. Nanoantioxidants that are reputed for their remarkable antioxidant features have been extensively reviewed [205] in a recent review by Khalil et al., 2020. These include SiO_2_ nanoparticles, Au NPs, Ag NPs, and Fe_2_O_3_ NPs, ceria nanoparticles that have proven to show enhanced antioxidant activity when functionalized with 2′,3,4′,5,7-pentahydroxyflavone, PEG, Polly tannic acid, salvianic acid, trolox, lignin, dextran, curcumin, and the like [205]. Such nanocomposites have been proven to be highly effective in various other fields, nanoantioxidants such as these should be combined with chitosan, to harness the full potential of such a nanocomposite. This is one area worth probing and expanding. Figure 3 projects the areas that need to be explored as future perspective in this subject area.

More focus on application of modified chitosan nanocarriers as well as chitosan composites towards anticancer research is essential. Not much has been achieved in this direction. This review expects expansion in these applications, via synthesis of such composite materials as well as their application into anticancer research. Not many clinical studies have been reported from the existing chitosan nanoformulations; this deserves research inputs.

## 7. Conclusions

This review highlights the importance of various modifications and functionalization on chitosan and nanochitosan. Given the fact that a biodegradable material, obtained from food waste, can be put to effective biomedical use, there is certainly plenty of room to improvise and expand the horizons. When resources are already running out, such recycled resources are essentially the future.

## Figures and Tables

**Figure 1 polymers-13-03138-f001:**
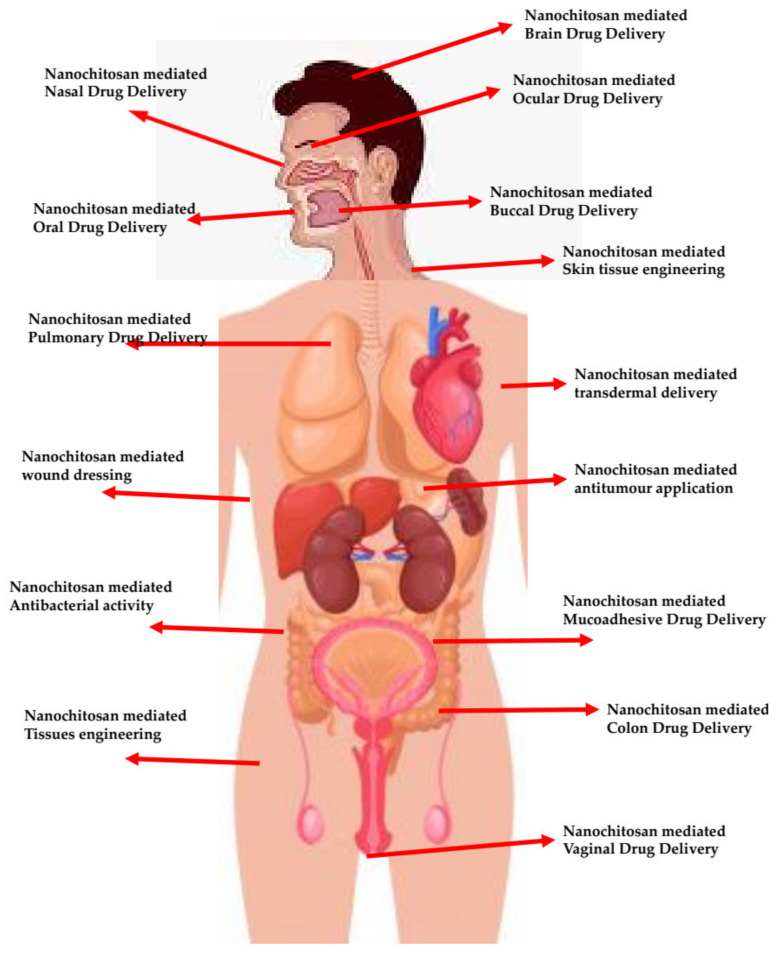
Overview of chitosan nanomaterial-based biomedical applications.

**Figure 2 polymers-13-03138-f002:**
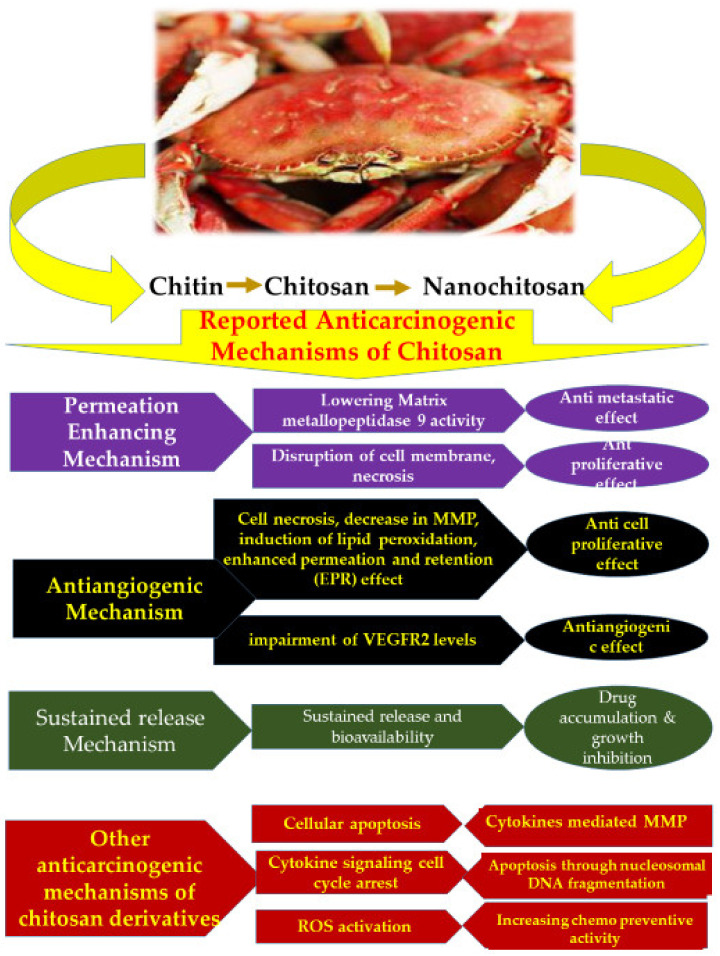
Overview of the anticarcinogenic mechanisms of chitosan and its forms that have been reported in literature.

**Figure 3 polymers-13-03138-f003:**
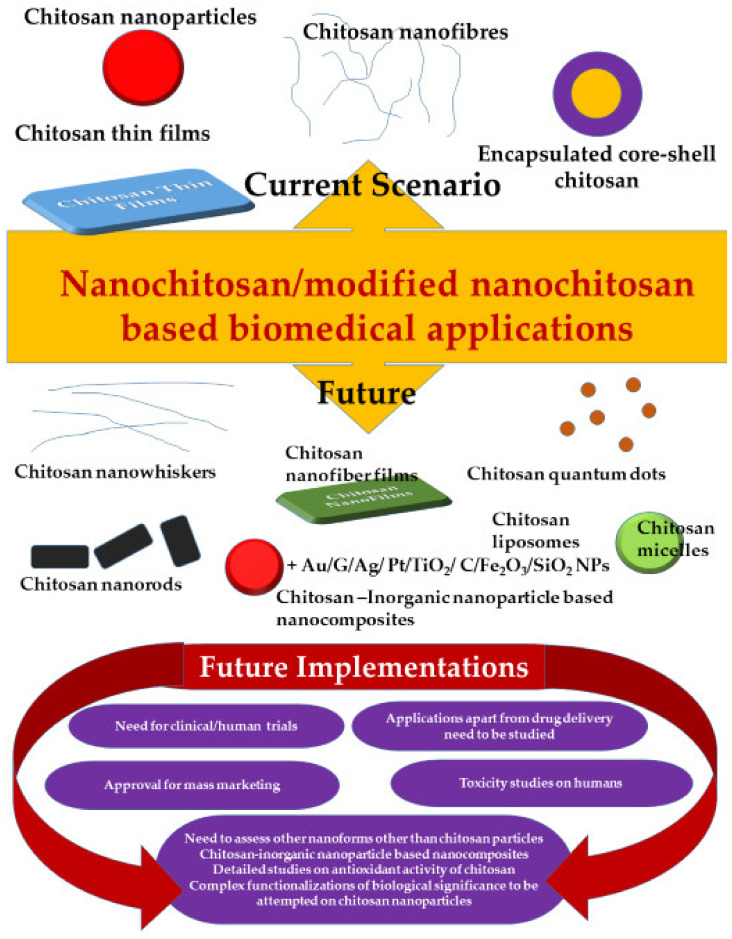
Future of chitosan nanomaterial based biomedical applications.

**Table 1 polymers-13-03138-t001:** Biomedical applications of modified chitosan nanomaterials.

Functionalized/Surface Modified/Encapsulated Chitosan Nanomaterial	Purpose	Application	Reference
Chitosan, soybean lecithin nanoparticles	Oral drug delivery	Intestinal permeation of tamoxifen through the rat intestinal wall	[68]
Chitosan nanoparticles	Oral drug delivery	Sunitinib drug delivery	[70]
LMW chitosan NP	Oral drug delivery	Solubility and bioavailability of Hydrophobic Bay41-4109 in rats	[72]
Chitosan, TPP	Oral delivery of insulin	Decreased glycaemia in diabetic rats after administering insulin-chitosan nanoparticles, in vivo	[71]
Chitosan HCl, Poloxamer 188, sodium glycolate, gelatin, soya lecithin	Oral delivery of Cyclosporin-A	Beagle dogs showed relative bioavailability of Cy-A was significantly increased by chitosan nanoparticles, in vivo.	[64]
Chitosan carboxymethyl chitosan	oral antigen delivery in fish vaccination	Extra cellular products (ECPs) of *Vibrio anguillarum*	[87]
Chitosan LMW, sodium tripolyphosphate (TPP), fluorenyl-methyloxycarbonyl chloride (FMOC)	Oral drug delivery	Chitosan nanoparticles released Alendronate sodium faster in 0.1 N HCl compared to PBS	[88]
Chitosan LMW, sodium tripolyphos-phate, tris[2-carboxyethyl] phosphine hydrochloride (TCEP)	Oral drug delivery	Enhanced intestinal absorption of catechins	[67]
Chitosan, STPP, sodium alginate	Oral drug delivery	Alginate coated chitosan nanoparticles containing enoxaparin for oral controlled release	[89]
Chitosan, deoxycholic acid, vitamin B12	Oral drug delivery	Enhancement of scutellarin oral delivery	[90]
Chitosan, Tc-methylene di-phosphonate	Oral drug delivery	Chitosan nanoparticles/F nanoparticles stable in the stomach and decompose in the intestine	[91]
Chitosan, PLGA, streptozotocin	Oral drug delivery of Tolbutamide	PLGA nanoparticles modified with chitosan to form TOL-CS-PLGA NPs to improve bioavailability and reduce dose frequency	[92]
Chitosan LMW, penta sodium tripolyphos-phate	Oral drug release	Gemcitabine-loaded chitosan nanoparticles (Gem- Chitosan nanoparticles) for oral bio-availability enhancement	[93]
Sodium alginate, chitosan, streptozotocin	Naringenin nanoparticles have better efficacy in lowering blood glucose levels compare to free drug	Alginate coated chitosan core shell nanoparticles for effective oral delivery	[94]
*N*-carboxymethyl chitosan, chitosan hydrochloride	Oral drug delivery	EGCG-chitosan/β-Lg NPs to achieve prolonged release during oral administration in gastrointestinal tract	[95]
Chitosan, sodium alginate, sodium pyruvate, l-glutamine	Oral drug delivery	Quercetin-chitosan/alginate nanoparticles high antioxidant property no systemic toxicity	[96]
Chitosan nanoparticles -TPP, lactose, Tween 80	Oral drug delivery	90% release of RFM from Chitosan nanoparticles within 24 h, in vitro	[75]
Hydroxypropyl-beta-cyclodextrin (HPβCD), mannitol, lactose, TPP, l-leucine	Pulmonary delivery	Chitosan nanoparticles for pulmonary delivery of itraconazole as a dry powder formulation	[76]
*N*,*N*,*N*-tri-methyl chitosan, TPP	Pulmonary drug delivery	Cellular uptake of Bac-TMC3/TPP/siRNA nananoparticles greatly enhanced by clathrin -mediated cellular uptake pathway	[97]
Chitosan, lipoid S100, glycol chitosan	Pulmonary drug delivery	LMWH chitosan and glycol Chitosan nanoparticles for enhancing pulmonary absorption of LMW heparin	[98]
Chitosan thioglycolic acid, TPP	Pulmonary drug delivery	Theophylline-thiolated Chitosan nanoparticles enhances theophylline’s capacity to alleviate allergic asthma	[99]
Thiolated chitosan	Pulmonary drug delivery	In vitro slow and sustained release of leuprolide from thiolated chitosan about 43% in 2 h	[100]
Chitosan, methylated β-cyclodextrin, TPP	Intranasal administration	Estradiol-chitosan nanoparticles for improving nasal absorption and brain targeting	[101]
LMW Chitosan, TPP, trehalose	Intranasal immunization	Tetanus toxoid chitosan nanoparticles (TT-CS NPs) as a new long-term nasal vaccine delivery vehicle	[102]
Chitosan, 4-CBS, TPP, 1-ethyl-3-(3-dimethylaminopropyl)carbodiimide HCl (EDAC)	mucoadhesive drug delivery	In vitro drug release of DOX loaded 4-CBS-chitosan/PLA nanoparticles showed sustained release up to 26 days	[103]
Chitosan (MW = 600 kDa), methane-sulfonic acid, oleoyl chloride, sodium bicarbonate, glycidyl-trimethyl ammonium chloride	oral administration with enhanced mucoadhesion	In vivo toxicology study was performed in zebrafish embryos	[104]
Chitosan, 1-ethyl-3-(3-dimethylaminopropyl) carbodiimide hydrochloride (EDC. HCl), *N*-hydroxyl succinimide	mucoadhesive drug delivery	Mucosal adhesion and drug release of cetirizine-chitosan	[105]
Chitosan, lactic acid	mucoadhesive drug delivery	Chitosan-based 5-ALA mucoadhesive film to enhance its retention in oral mucosa	[106]
Chitosan Low, polycaprolactone, glycerol nanoparticle	mucoadhesive drug delivery	Mucoadhesive films containing curcumin-loaded nanoparticles to prolong the residence time in the oral cavity and to increase drug absorption through the buccal mucosa	[107]
Chitosan, TPP, Carbopol 940, poloxamer 407	Drug delivery	Propranolol-chitosan nanoparticles of transdermal gels to improve the systemic bioavailability of the drug	[108]
Resomer PLGA, ploxamer 188, sorbitan monoaleate, chitosan	flavonoid enriched cytotoxic film	EFF-Cg nanocomposites chitosan film containing PLGA NPs, showed low toxicity	[109]
Chitosan, TPP, Triton X-100	Oral drug delivery of alginate and pectin	Preparation of alginate and pectin chitosan nanoparticles for oral drug delivery	[110]
Chitosan MMW, PEG, PVP, trehalose	Insulin release	Chitosan films with insulin loaded PEG-b-PLA nanoparticles with sustained release	[111]
Chitosan buccal films of insulin loaded poly (ethylene glycol) methyl ether-block-polylactide (PEG-b-PLA) NP	Insulin release	Excellent mucoadhesive properties and insulin release	[77]
Polycaprolactone nanoparticles coated with chitosan	Buccal delivery	Delivery of curcumin	[57,78]
EFF-Cg loaded PLGA nanoparticles as chitosan films.	Buccal delivery	The bioavailability of EFF-Cg was improved and no signs of cytotoxicity were seen	[78]
Conjugating C2-N position of chitosan with aromatic sulfonamide, 4-carboxybenzenesulfonamide-chitosan (4-CBS-chitosan)	drug release in small intestine	Mucoadhesive property of chitosan in stomach acidic environment increased	[57]
Entrapping ovalbumin (OVA) into Eudragit S 100, trimethylchitosan, PLGA, PEG-PLGA and PEG-PCL	inflamed colon drug delivery	Nanoparticles with trimethyl chitosan have shown the highest permeability of OVA. And high permeability	[84]
chitosan-carboxymethyl starch nanoparticles of 5-aminosalicylic acid	Drug delivery for inflammatory bowel disease	Controlled drug release	[85]
Rosmarinic acid loaded chitosan nanoparticles	ocular delivery	The nanoparticles showed no cytotoxicity against the retinal pigment epithelium nor the human cornea cell line.	[112]
Chitosan coated PCL nanocapsules embedded in hydroethylcellulose gel	vaginal delivery to treat human papillomavirus infection	Imiquimod formulated chitosan coated PCL nanocapsules embedded in hydroethylcellulose gel	[86]
PCL nanocapsules embedded in chitosan hydrogel	vaginal delivery to treat human papillomavirus infection	Imiquimod delivery to vagina	[113]
Limonene coated in chitosan	Enhancing antioxidant activity	Limonene-chitosan encapsulation has antioxidant activity with IC50 value of 116 ppm	[114]
Carboxymethyl chitosan nanofibres PEO and PVA-Ag	Biomedical application	Antibacterial	[115]
Carboxymethyl chitosan nanofibres—PVA\PVA\silk fibroin	Biomedical application	Wound dressing	[116]
Quaternized chitosan nanofibres-coPLA/DOX/PLA	Biomedical application	Antitumor	[117]
Quaternized chitosan nanofibres—PVA/PVP	Biomedical application	Antibacterial	[118,119,120]
Quaternized chitosan nanofibres—graphene	Biomedical application	Virus removal	[121]
Quaternized chitosan nanofibres—PLA	Biomedical application	Wound dressing	[122]
Poly-3-caprolactonegra chitosan nanofibres	Biomedical application	Skin tissue engineering	[123]
Chitosan/Albumin Nanoparticles	Drug delivery	Used as a hydrophobic drug nanocarrier in pharmaceutical and medical applications	[124]
Chitosan/Curcumin nanoparticles	Drug delivery	Transdermal delivery	[125]
Chitosan/Sodium Nitrate nanoparticle	Drug delivery	Delivery of DOX	[103]
Chitosan/HA nanoparticle	Drug encapsulation	Used to encapsulate a chemotherapeutic drug	[126]
Chitosan/Paromomycin nanoparticle	Anti leishmaniasis	Treatment of leishmaniasis, especially when the current drugs are impaired by resistance	[127]
Chitosan/Lipid Hybrid nanoparticles	Drug delivery	Controlled delivery of cisplatin	[128]
Chitosan/Human serum albumin nanoparticle	Drug delivery	Nose-to-brain drug delivery	[129]
Chitosan/Polylactide nanoparticle	Drug delivery	Delivery of therapeutics for triple-negative breast cancer treatment	[130]
Chitosan/Cadmium Quantum Dots	Drug delivery	Drug delivery of Sesamol	[131]
Chitosan/Silica Nanoparticles Thin Film	Drug delivery	DOX delivery	[132]
Chitosan/PVA nanoparticle	Oral delivery	Sustained release of the immunosuppressant drug mycophenolate mofetil	[133]
Chitosan-carbon dot hybrid nanogel	Anticancer activity	Photothermal therapy-chemo	[134]
PEGylated and fluorinated chitosan nanogel	Drug delivery	Targeted drug delivery	[135]
Chitosan grafted MPEG-PCL micelles	Drug delivery	Ocular delivery of hydrophobic drug	[136]
Arginine-modified nanostructured lipid carriers	Drug delivery	Anticancer drug delivery	[137]
Glycosaminoglycan modified chitosan liposome	Drug delivery	Antimalarial	[138]
Gold nanoshell-coated liposomes	Anticancer	Photothermal and chemotherapy	[139]
Glycol chitosan-coated liposomes	Drug delivery	pH-responsive drug-delivery	[140]
Chitosan nanoparticles-doped cellulose films	Antibacterial activity	Inhibition of Escherichia coli	[141]

**Table 2 polymers-13-03138-t002:** Established mechanisms for anticarcinogenic activity of chitosan and nanochitosan.

Test	Chitosan Form	Target Cell Line	Mode of Action	Reference
In vitro and in vivo	Chitosan	MDA-MB-231	Permeation enhancement, lowering of matrix metallopeptidase 9 activity leading to antimetastatic effect	[180,181]
In vitro	Chitosan	T24 urinary bladder cell lines	Disruption of cell membrane, necrosis resulting in antiproliferative effect	[182,183]
In vitro	Chitosan nano particles	Human hepato carcinoma	Antiangiogenic effect through, antiangiogenic action of chitosan nanoparticles and impairment of vascular endothelial growth factor (VEGFR) 2 levels	[184,185]
In vitro	Chitosan nano particles	BEL7402, HT-29	Cell necrosis, lipid peroxidation, decrease in MMP, enhanced permeation and retention (EPR) effect, resulting in inhibition of cellular proliferation	[155,186,187,188,189]
In vivo	Mifepristone (MIF) loaded chitosan nano particles	Solid tumor	Sustained release and enhancement of bioavailability of drug. Drug accumulation and growth inhibition	[190]

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
