# Peer review of "Evaluating the Anticarcinogenic Activity of Surface Modified/Functionalized Nanochitosan: The Emerging Trends and Endeavors"

_polymers, 2021, doi:10.3390/polym13183138_

Round 1
Reviewer 1 Report
Review of low quality and low novelty. It needs much improvement to be suitable for publication.
Author Response
Review of low quality and low novelty. It needs much improvement to be suitable for publication.
Ans. We thank the Editor and the reviewers team for their time and meticulous review. We have revised the manuscript as per the three reviewers suggestions.
Thank you
Reviewer 2 Report
The present review on chitosan activity as anticarcinogenic requires first of all moderate English revision, figures revision in terms of readable font and higher resolution, respectively references arrangement.
Abstract: Please revise "ideal functionalization’s", the "s" is not necessary;
Introduction, first paragraph: "The degree of deacetylation of chitosan refers to the molar fraction of N-acetylated units (DA) or percentage of acetylation (DA%)." The degree of deacetylation (DDA) is opposed to the degree of acetylation (DA), DA=100-DDA or DDA=100-DA, please revise. DA is also very important for solubility and in separating the terms chitin-chitosan: if DA>50% the biopolymer is chitin-like, if DA<50% the biopolymer is chitosan-like. Please document this and the physical-chemical properties correlated with DA, and particularly for the human physiological fluids to serve the purpose of the paper.
Second paragraph: "Its amino groups have primary and secondary hydroxyl groups at the C2-, C3-, and C6- positions." The hydroxyls are not covalently bounded by amino groups; please correct and revise the non-covalent interactions in chitosan.
Page 2: "Aranaz et al., reported that when the DDA increases, the solubility of chitosan also increases and bio interactions increase [3, 29, 30]." Aranaz et al. did not wrote ref [29,30], please revise the citation style;
Please correct "Habstraction reaction" to "H-abstraction reaction";
"They concluded that higher DDA and lower MW are characteristic patterns for higher antioxidant activity [40]" - please give the values also;
Page 4: "...semi-interpenetrating polymer (semi-IPN) [50]". IPN probably comes from "interpenetrating polymer network", please revise;
Page 6: "Chitosan-vancomycin nanoparticles for colon delivery were prepared for better drug release [89]. Coco et al. have studied the comparative ability of chitosan nanoparticles against other polymers, for inflamed colon drug delivery [90]." The reference numbers are not the corresponding ones, please revise;
Rearrange Table 2 according to the style of Polymers Journal and the columns width;
Repair Fig.2 since the crab is way too flat, increase font text and figure resolution;
6.3. "Unexplored arenas" or "Unexplored areas"?
Fig.3 should be placed above the last paragraph and the smallest text increased, and resolution also;
Many references have unnecessary numbers and symbols, please use a reference program (EndNote or others); for example ref [178,179]. Ref [216] is displaced and of different font.
Author Response
The present review on chitosan activity as anticarcinogenic requires first of all moderate English revision, figures revision in terms of readable font and higher resolution, respectively references arrangement.
Ans. We have revised the manuscript based on your valuable suggestions. thank you for your patience and for all your inputs. We have revised the language, and figures and references
Abstract: Please revise "ideal functionalization’s", the "s" is not necessary;
ans. Revised
Introduction, first paragraph: "The degree of deacetylation of chitosan refers to the molar fraction of N-acetylated units (DA) or percentage of acetylation (DA%)." The degree of deacetylation (DDA) is opposed to the degree of acetylation (DA), DA=100-DDA or DDA=100-DA, please revise. DA is also very important for solubility and in separating the terms chitin-chitosan: if DA>50% the biopolymer is chitin-like, if DA<50% the biopolymer is chitosan-like. Please document this and the physical-chemical properties correlated with DA, and particularly for the human physiological fluids to serve the purpose of the paper.
Ans. Documented
Second paragraph: "Its amino groups have primary and secondary hydroxyl groups at the C2-, C3-, and C6- positions." The hydroxyls are not covalently bounded by amino groups; please correct and revise the non-covalent interactions in chitosan.
Ans. Corrected. thank you
Page 2: "Aranaz et al., reported that when the DDA increases, the solubility of chitosan also increases and bio interactions increase [3, 29, 30]." Aranaz et al. did not wrote ref [29,30], please revise the citation style;
Ans. Sorry about that, corrected.
Please correct "Habstraction reaction" to "H-abstraction reaction";
Ans. Corrected.
"They concluded that higher DDA and lower MW are characteristic patterns for higher antioxidant activity [40]" - please give the values also;
Ans. Values not specified.
Page 4: "...semi-interpenetrating polymer (semi-IPN) [50]". IPN probably comes from "interpenetrating polymer network", please revise;
Ans. revised as semi- interpenetrating polymer network that what semi-IPN stands for. Thankyou.
Page 6: "Chitosan-vancomycin nanoparticles for colon delivery were prepared for better drug release [89]. Coco et al. have studied the comparative ability of chitosan nanoparticles against other polymers, for inflamed colon drug delivery [90]." The reference numbers are not the corresponding ones, please revise;
Ans. Revised
Rearrange Table 2 according to the style of Polymers Journal and the columns width;
Ans. Rearranged
Repair Fig.2 since the crab is way too flat, increase font text and figure resolution;
Ans. Repaired
6.3. "Unexplored arenas" or "Unexplored areas"?
ans. Changed
Fig.3 should be placed above the last paragraph and the smallest text increased, and resolution also;
Ans. Yes we have carried out these corrections.
Many references have unnecessary numbers and symbols, please use a reference program (EndNote or others); for example ref [178,179]. Ref [216] is displaced and of different font.
Ans. We have checked on the references. thank you.
Reviewer 3 Report
Its is interesting review. I believe that this manuscript can ultimately get acceptance on the condition of rigorous revision addressing comments shown below:
- Its would better if authors keep either modified or functionalized term in the title of article.
- Authors need to cite recent literatures and put more figures from literature in manuscript.
- This manuscript is well organized but lack of specific comparative analysis. What are the advantages of nanochitosan compared with traditional nanotechnology?
- Section 3 and section 4 heading seems similar. Authors need to check it carefully. At same time please elaborate section 3 more by citing some recent literatures.
- It would be better if authors combine section 3 and 4 together and make separate heading for 4.1.
- The authors should summarize the current approaches to fabricate nanochitosan and compare their advantages and disadvantages in order to draw the reader's attention.
- In conclusions and perspectives, the author should consider giving some methodological design about how to improve the performance of naochitosan.
- There are too many grammatical mistakes, try to remove these mistakes before it published.
Author Response
Its is interesting review. I believe that this manuscript can ultimately get acceptance on the condition of rigorous revision addressing comments shown below:
Ans. We thank the reviewer for the appreciation. We are greatly encouraged. We have now revised the review according to your valuable comments.
- Its would better if authors keep either modified or functionalized term in the title of article.Ans. Actually chitosan modifications include both, surface modifications as well as functionalization, that is why we specified both. thank you
- Authors need to cite recent literatures and put more figures from literature in manuscript. Ans. cited. We have three figures currently, we decided to go with the figures we have, since literature based figures will involve delays in acquiring permission. Thank you for your understanding.
- This manuscript is well organized but lack of specific comparative analysis. What are the advantages of nanochitosan compared with traditional nanotechnology?Ans. Thank you. The advantages of nanochitosan, is the very fact that this is biodegradable material, unlike the other synthesized non-ecofriendly nanoparticles prepared by traditional nanotechnological processes.
- Section 3 and section 4 heading seems similar. Authors need to check it carefully. At same time please elaborate section 3 more by citing some recent literatures.Ans. Section 3 is about the modifications reported for chitosan and section 4 is about the modifications on nanochitosan. Section 3 is merely a snap shot of the various modifications involved. We focus more on the nanochitosan modifications. Thank you for your kind understanding.
- It would be better if authors combine section 3 and 4 together and make separate heading for 4.1Ans. As explained above, we try to restrict on section 3 and elaborate on section 4. Thats why we have divided these sections likewise. thank you.
- The authors should summarize the current approaches to fabricate nanochitosan and compare their advantages and disadvantages in order to draw the reader's attentionAns. Fabrication methods have been so well reviewed and reported and there are excellent reviews on this subject area. That is why we have stuck to the theme of the review. thank you for your understanding.
- In conclusions and perspectives, the author should consider giving some methodological design about how to improve the performance of naochitosan - Ans. We have revised the conclusions.
- There are too many grammatical mistakes, try to remove these mistakes before it published. Ans. We apologize for this inconvenience. We have now had the manuscript read and revised by a native english speaker. the manuscript is now rid of language issues. Thank you.
Round 2
Reviewer 1 Report
The authors addressed well the reviewers' comments and suggestions.
Reviewer 3 Report
Accepted in present form
This manuscript is a resubmission of an earlier submission. The following is a list of the peer review reports and author responses from that submission.
Round 1
Reviewer 1 Report
The article is another review on a subject addressed by some of the Authors in other two reviews that are not cited, although some paragraphs are identical. The previews reviews are both from this year:
10.3390/ nano11030821
10.3390/polym13142256
The paragraphs identical with the ones from previous articles should be revised, for example:
Page 4: "The most common methods for obtaining chitosan nanomaterials are though: Ionotropic gelation, microemulsion, emulsification solvent diffusion and emulsion based solvent evaporation. Each of these methods influence the particle size and surface charge of nanochitosan and impact the molecular weight and degree of acetylation."
Please explain the term "vibes" from the title, and the motivation, since it is used only in the title.
Abstract L2: What is the relevance of "nutraceutical" related to "anticarcinogenic"?
The writing style is also full of mistakes, for example:
- inappropriate punctuation, like comma between subject and predicate starting from the Abstract: "Chitosan, begins...", or between "The biopolysaccharide, chitosan..."
- "Chitosan contains three reactive groups at C-2, C-3, and C-6 positions, respectively [5]. Among these three, the amine at the C-2″..." What is the relation between C-2, C-3, C-6 and C-2"?
- "The degree of deacetylation (DDA) has an important say on the bioactivity of chitosan, as they are directly related..." Who are "they"?
- Introduction, 3rd paragraph: "pharmaceutical arenas"
"High scientific interest and research has been roused..."
Please use a similitude software to proof-check the article, and revise the English.
Author Response
Reviewer 1
We would like to thank our Editor and Reviewers for this opportunity to revise our manuscript. Thank you for patiently reviewing our manuscript and giving us your valuable suggestions, amidst you’re your own schedules. We greatly appreciate this gesture. We have now modified our manuscript according to your suggestions to the best possible. We have also given a point by point response to your queries below. We thank you for your time and inputs.
The article is another review on a subject addressed by some of the Authors in other two reviews that are not cited, although some paragraphs are identical. The previews reviews are both from this year:
10.3390/ nano11030821
10.3390/polym13142256
Ans. We do understand your concern, the papers you have mentioned are both from our group, 10.3390/ nano11030821-Nanochitosan: Commemorating the Metamorphosis of an ExoSkeletal Waste to a Versatile Nutraceutical; 10.3390/polym13142256-Green Synthesized Chitosan/Chitosan Nanoforms/Nanocomposites for Drug Delivery Applications. The first paper deals with touching on the nutraceutical aspects of chitosan nanomaterials. The second paper deals with reviewing the progress of green synthesized chitosan and its nanoforms for drug delivery applications. The current paper focusses on the various advances made through surface modified or functionalized chitosan materials and their anticancer applications. Thus, all three are different. We apologize that there are similar textual overlaps. Will replace them. Thank you for your patience.
The paragraphs identical with the ones from previous articles should be revised, for example:
Page 4: "The most common methods for obtaining chitosan nanomaterials are though: Ionotropic gelation, microemulsion, emulsification solvent diffusion and emulsion based solvent evaporation. Each of these methods influence the particle size and surface charge of nanochitosan and impact the molecular weight and degree of acetylation."
Ans. Revised. Thank you.
Please explain the term "vibes" from the title, and the motivation, since it is used only in the title.
Ans. Vibes are effects, since you are not comfortable with it, we have removed it in the revision. Thank you
Abstract L2: What is the relevance of "nutraceutical" related to "anticarcinogenic"?
Ans. A nutraceutical is defined as any substance that is a food or part of a food and provides medical or health benefits. Chitosan is part of a food and has medicinal applications and hence is a nutraceutical. And chitosan also exhibits anticancer activities. Thank you.
The writing style is also full of mistakes, for example:
- inappropriate punctuation, like comma between subject and predicate starting from the Abstract: "Chitosan, begins...", or between "The biopolysaccharide, chitosan..."
Ans. Revised. Very sorry about these.
- "Chitosan contains three reactive groups at C-2, C-3, and C-6 positions, respectively [5]. Among these three, the amine at the C-2″..." What is the relation between C-2, C-3, C-6 and C-2"?
Ans. C2 is the position that plays a key role in biological activities as well as in surface modifications that influence biological activities. I think this doubt has risen because ewe have loosely described this. We have revised and modified this description. Thank you.
- "The degree of deacetylation (DDA) has an important say on the bioactivity of chitosan, as they are directly related..." Who are "they"?
Ans. Typo, language issue, revised. Sorry.
- Introduction, 3rd paragraph: "pharmaceutical arenas"
"High scientific interest and research has been roused..."
Please use a similitude software to proof-check the article, and revise the English.
Ans. We are very sorry about the language issues in the manuscript. We now have had a native English speaker to check and edit the entire manuscript, thank you for your patience.
Reviewer 2 Report
The review entitled “Evaluating the anticarcinogenic vibes of surface
modi-fied/functionalized nanochitosan: the emerging trends and endeavors” is interesting. However, it presents a quite poor approach and many changes need to be addressed in order to be suitable for publication:
1)Many more figures are necessary in a review
2)Also, more tables are necessary such as a Table with all the methods developed so far to produce chitosane nanoparticles and another table with chitosan nanoparticles used as nanocarriers
3)The toxicological aspects of chitosane used as nanocarrier in each study should be mentioned. The toxicological aspects commented in the last sentence could be organized in a new section entitled Toxicological aspects or something like this
4)The anticarcinogenic mechanisms should be discussed in detail
5)The structure of the review is difficult to follow. It must be improved
6) Other specific applications of chitosan-based nanoparticles such as the management of fungal diseases should be commented in the review (https://doi.org/10.3390/app11157119)
7)There is a sentence in bold in the last sentence
Author Response
The review entitled “Evaluating the anticarcinogenic vibes of surface
modi-fied/functionalized nanochitosan: the emerging trends and endeavors” is interesting. However, it presents a quite poor approach and many changes need to be addressed in order to be suitable for publication:
Ans. We thank you for your valuable comments and inputs into our manuscript. We Have revised the manuscript as per your suggestions as is possible. We have responded point by point to your queries below. Thank you.
1)Many more figures are necessary in a review
Ans. Actually, we have two figures in this review. Chitosan manuscripts are many and figures in those manuscripts, have covered many areas. Our figures are plotted unique to the specific topics addressed. Hope you will understand our reasoning. Thank you.
2)Also, more tables are necessary such as a Table with all the methods developed so far to produce chitosane nanoparticles and another table with chitosan nanoparticles used as nanocarriers
Ans. In terms of tables, methods, applications of chitosan, chitosan nanoparticles are also represented by a multitude of tables in publication. Here we present a elaborate table, one long table on the specific topic addressed in this manuscript. As such the Table runs long. And the manuscript is also long 13k words. That is why we have limited from repetitions.
3)The toxicological aspects of chitosane used as nanocarrier in each study should be mentioned. The toxicological aspects commented in the last sentence could be organized in a new section entitled Toxicological aspects or something like this
Ans. The toxicological aspects of chitosan have been addressed in our previous reviews as well as other previous reviews. That is not within the focus of this review topic, since you have suggested, we have organized the information in a section. Thank you.
4)The anticarcinogenic mechanisms should be discussed in detail
Ans. Described in the revision and added a citation, Hari Sharan Adhikari, Paras Nath Yadav, "Anticancer Activity of Chitosan, Chitosan Derivatives, and Their Mechanism of Action", International Journal of Biomaterials, vol. 2018, Article ID 2952085, 29 pages, 2018. Thank you.
5)The structure of the review is difficult to follow. It must be improved
Ans. Modified a bit thank you.
6) Other specific applications of chitosan-based nanoparticles such as the management of fungal diseases should be commented in the review (https://doi.org/10.3390/app11157119)
Ans. This would be deviating, not linked with the current manuscript. Thank you.
7)There is a sentence in bold in the last sentence
Ans. Revised. Thank you.
Round 2
Reviewer 1 Report
The Authors revised the manuscript.
Reviewer 2 Report
The review is interesting but needs much more work to be suitable for publication in a high IF journal such as Polymers. It needs to be improved in many aspects such as the inclusion of more descriptive figures and tables. The toxicological aspects of chitosan used as a nanocarrier in each study should be approached in detail in a separated section. The anticarcinogenic mechanisms should be discussed in detail. The structure of the review is difficult to follow.